# Hess Opinions: An interdisciplinary research agenda to explore the unintended consequences of structural flood protection

Giuliano Di Baldassarre[1,2], Heidi Kreibich[3], Sergiy Vorogushyn[3], Jeroen Aerts[4], Karsten Arnbjerg-Nielsen[5], Marlies Barendrecht[6], Paul Bates[7], Marco Borga[8], Wouter Botzen[4,9], Philip Bubeck[10], Bruna De Marchi[11], Carmen Llasat[12], Maurizio Mazzoleni[13], Daniela Molinari[14], Elena Mondino[1,2], Johanna Mård[1,2], Olga Petrucci[15], Anna Scolobig[16], Alberto Viglione[17], and Philip J. Ward[4]

[1]Department of Earth Sciences, Uppsala University, Uppsala, 75236, Sweden
[2]Centre of Natural Hazards and Disaster Science (CNDS), Sweden
[3]GFZ German Research Centre for Geosciences, Potsdam, 14473, Germany
[4]Institute for Environmental Studies, Vrije Universiteit Amsterdam, Amsterdam, 1081, The Netherlands
[5]Department of Environmental Engineering, Technical University of Denmark, Kgs. Lyngby, 2800, Denmark
[6]Centre for Water Resource Systems, Vienna University of Technology, Vienna, A-1040, Austria
[7]School of Geographical Sciences, University of Bristol, Bristol, BS8 1SS, UK
[8]Department of Land, Environment, Agriculture and Forestry, Università degli Studi di Padova, Padova, 35122, Italy
[9]Utrecht University School of Economics (USE.), Utrecht University, Utrecht, The Netherlands.
[10]Institute of Earth and Environmental Science, University of Potsdam, Potsdam, 14469, Germany
[11]SVT, Centre for the Study of the Sciences and the Humanities, University of Bergen, Bergen, 5020, Norway
[12]Department of Applied Physics, University of Barcelona, Barcelona, 08007, Spain
[13]Department of Integrated Water Systems and Governance, IHE Delft, Delft, 2601, The Netherlands
[14]Department of Civil and Environmental Engineering, Politecnico di Milano, Milan, 20133, Italy
[15]CNR-IRPI National Research Council - Research Institute for Geo-Hydrological Protection, Rende (CS), 87036, Italy
[16]Department of Environmental Systems Science, ETH Zürich, Zürich, 8092, Switzerland
[17]Centre for Water Resource Systems, Vienna University of Technology, Vienna, A-1040, Austria

*Correspondence to*: Giuliano Di Baldassarre (giuliano.dibaldassarre@geo.uu.se)

**Abstract.** One common approach to cope with floods is the implementation of structural flood protection measures, such as levees or flood-control reservoirs, which substantially reduce the probability of flooding at the time of implementation. Numerous scholars have problematized this approach. They have shown that increasing the levels of flood protection can attract more settlements and high-value assets in the areas protected by the new measures. Other studies have explored how structural measures can generate a sense of complacency, which can act to reduce preparedness. These paradoxical risk changes have been described as 'levee effect', 'safe development paradox' or 'safety dilemma'. In this commentary, we briefly review this phenomenon by critically analysing the intended benefits and unintended effects of structural flood protection, and then we propose an interdisciplinary research agenda to uncover these paradoxical dynamics of risk.

## 1 Premise

Economic losses caused by floods are increasing in many regions of the world, and flood risk will likely further increase because of climatic and socioeconomic changes (Aerts et al., 2014; Alfieri et al., 2016). One common approach to cope with floods is the implementation of structural flood protection measures, such as levees or flood-control reservoirs. These types of infrastructure have been implemented for many centuries in different areas around the world, as they can significantly

reduce the probability of flooding. In the Netherlands, for example, the current levee system is able to withstand floods up to return periods ranging from 500 to 10,000 years (De Moel et al., 2011). In many parts of Europe, USA and Australia, flood protection measures are typically designed to protect people and assets from events with return periods between 100 and 1,000 years (Bubeck et al. 2017). Conversely, most low-income countries currently have lower protection standards

(Scussolini et al., 2016), and flooding events are therefore more frequent.

Recently, a global study of flood risk in a changing climate (Ward et al., 2017) has shown that the expected benefits of structural protection measures preventing frequent flooding often outweigh their building costs. This study made the (common) assumption that future flood exposure depends on socioeconomic trends only, and not on the level of flood protection. However, since the studies of Gilbert White about human adjustments to floods (White, 1945), numerous

scholars (White, 1994; Tobin, 1995; Burby, 2006; Kates et al., 2006; Burton and Cutter, 2008; Montz and Tobin, 2008; Scolobig and De Marchi, 2009; Ludy and Kondolf, 2012; Di Baldassarre et al., 2013ab; 2015; Wenger, 2015) have shown that increasing levels of flood protection can also be associated with unexpected increases in flood exposure. Fig. 1 depicts how the urbanisation of flood-prone areas (and therefore flood exposure) can be influenced by structural flood protection. The left and the right panel of Fig. 1 start from the same historical settlement (i.e. the orange buildings), and then show the

urbanisation of flood-prone areas. If such an urbanisation was triggered by socioeconomic trends only (e.g. population growth), the spatial distribution of the new settlements would be the same. However, the presence of structural flood protection tends to create incentives to build closer to the river and therefore increases flood exposure (compare left and right panels of Fig 1). Thus, socioeconomic trends determine the amount of urbanisation increase, while the presence of structural flood protection influences the spatial location of new settlements and as such may lead to increased flood exposure. This

tendency is typically described as the 'levee effect', although some scholars have used different terms, such as 'safe development paradox' or 'safety dilemma' (Burby, 2006; Scolobig and De Marchi, 2009). This phenomenon can offset part of the intended benefits of structural flood protection and, paradoxically, flood risk can even increase in the medium-long term after the introduction or reinforcement of a structural flood protection (Kates et al., 2006; Montz and Tobin, 2008; Di Baldassarre et al., 2013b).

**2 The troubles with structural flood protection**

**2.1 Increasing exposure**

The aforementioned studies have discussed how building levees (or other types of structural protection measures, such as flood-control reservoirs) is often associated with more intense urbanization of flood-prone areas behind the levee (Fig. 1), i.e. more people and assets will eventually be exposed to less frequent, but potentially catastrophic flooding (Merz et al.,

2009). This phenomenon has been observed in many parts of the world, including: Bangladesh (Ferdous et al., 2018) in Asia, The Netherlands (De Moel et al., 2011), Central Pyrenees (Benito et al., 1998) and the Po River valley (Di Baldassarre et al., 2013b) in Europe; Brisbane (Bohensky and Leitch, 2014) in Australia; the Sacramento Valley (Ludy and Kondolf, 2012) and New Orleans (Colten, 2005; Kates et al., 2006; Colten and De Marchi, 2009) in the United States.

De Moel et al. (2011), for example, analysed changes in flood exposure in The Netherlands by using land-use data with information about the maximum flood inundation. The study showed that the urban area that can be potentially flooded has increased six-fold during the 20th century. Moreover, it showed that while the proportion of urban area in flood-prone areas substantially dropped after the occurrence of a catastrophic flooding in 1953, this proportion has started to grow again over the past decades (from about 27% to about 31%), as flood protection was increased by introducing numerous structural measures, such as the Delta Works. This growth has brought economic benefits to these areas, but also offset part of the decline in flood risk that resulted from the strengthening of flood protection.

It should be mentioned that urban growth behind the dikes is often factored into the risk analysis. A recent study (Hallegatte, 2017) finds that whilst structural protection measures can increase potential losses (especially of large events) due to increased exposure, it can also generate benefits through more investment and economic activity. Indeed, this is one of the goals of flood protection investments: not only to reduce flood risk, but also to make it possible to facilitate economic growth in areas that are flood-prone but valuable, e.g. coastal areas that offer low trade and transport costs or areas in cities that benefit from the proximity of jobs and services (Hallegatte, 2017). However, in other cases, urban growth in flood-prone areas goes beyond original plans, as depicted for example in Fig. 1, potentially leading to unforeseen increase in flood risk. If this happens does not dependent on the level of protection, but on risk communication and the specific societal and political context. In recent decades, it has been increasingly recognised in many countries that a residual risk of flooding remains behind levees (Bubeck et al. 2017; Penning-Rowsell et al., 2006). Before, structural flood protection was commonly accompanied by the belief that protected areas are save and flood problems are solved by means of engineering. The resulting increase in flood risk due to increased exposure. This can imply that, based on cost-benefit analysis (Kind, 2014), it becomes economically beneficial to strengthen flood protection again (see next Section 2.2). Thus, the overall impacts of the levee effect on urban growth and flood risk depend on the specific context in which levees are planned and designed.

## 2.2 Vicious cycles, lock-in conditions and unexpected failures

The levee effect can lead to self-reinforcing feedbacks: increasing protection levels favours intense urbanisation of floodplains that will then plausibly require even higher protection standards, as seen e.g. in The Netherlands (Di Baldassarre et al., 2015). Thus, it can generate lock-in conditions towards exceptionally high levels of flood protection and extremely urbanised floodplains. This lock-in condition can be unsustainable (e.g the maintenance of large infrastructure requires commitment of regular resources) or undesirable (e.g. large infrastructure can contribute to unfair distributions of risk; Masozera et al., 2007; Di Baldassarre et al., 2013b; Ferdous et al., 2018). Indeed, the costs and benefits of flood protection measures, as well as potential flood losses, are often not fairly shared across social groups (Kind et al., 2017), as seen in the aftermath of the catastrophic 2005 flooding of New Orleans (Kates et al., 2006).

Moreover, changes in technical flood protection inevitably cause spatial risk redistribution due to hydraulic interactions, e.g. risk shifts downstream due to increased levee heights upstream, but to date these effects remain poorly understood

(Vorogushyn et al., 2018). Similarly, there are reports of "levee wars", i.e. where local districts (or land owners) build higher levees to prevent local flooding and make other areas riskier (e.g. Allan James and Singer, 2008).

Lastly, the shift from frequent to rare-but-catastrophic flooding generated by structural flood protection causes serious problems for decision making in flood risk management, due to high uncertainty associated with the estimation of low probability flood events, such as the 1-in-100-year flood (Merz and Thieken 2005; Merz et al. 2009). Additionally, rare-but-catastrophic events bear the potential of unexpected negative consequences, as they can take society by surprise and lead to a complex web of socio-economic interactions (Di Baldassarre et al., 2016), perhaps beyond the recovery potential (Merz et al., 2015).

## 2.3 Increasing vulnerabilities

Increasing the levels of flood protection can also generate a sense of complacency among the protected people, which can reduce preparedness, thereby increasing vulnerability (Tobin, 1995). This additional facet of the levee effect was explored by Scolobig and De Marchi (2009) and De Marchi and Scolobig (2011) with reference to four communities in North Eastern Italy. Interviews, focus group discussions and surveys in these areas showed that residents of communities exposed to flood risk tend to underestimate, minimize or even neglect risk (see also the report in De Marchi et al., 2007). These studies showed that an important component of such an attitude is the false sense of security induced by the presence of (often impressive) structural works designed to limit risk and prevent damage. Apparently, the symbolic messages encrypted in stones ("no problem") are more powerful than the verbal messages conveyed in information campaigns ("you are protected, but not totally safe"). More specifically, De Marchi et al. (2007) report the level of agreement of the informed respondents with four statements about protection works gauged on a Likert scale from 1 to 5 (a response of 1 signifies strong disagreement with the statement, while a response of 5 indicates strong agreement). The statements are listed here from highest to lowest mean values:

- The protection works give a feeling of safety to the people living in the village (4.49).
- The protection works eliminate the possibility of serious damage (3.92).
- The protection works promote/help the economic development of the community (3.48).
- The protection works are too expensive compared to the expected benefits (1.76).

The high mean value (4.49 out of 5) relating to the first statement suggests that structural protection plays a role in inducing a feeling of safety among residents in these risky areas. Moreover, the high agreement (3.92) with the item "elimination of serious damage", indicated that there was very little awareness of residual risk. Thus, in this area, people protected by levees were not well motivated to undertake private precautionary measures and as such are more vulnerable towards flooding, as also found in Ludy and Kondolf (2012) in the Sacramento valley.

Yet, the reality is much more complex, as multiple factors drive risk perception and the adoption of protection measures. This leads to dissimilar outcomes in different contexts. For example, Botzen et al. (2009) found that people in The Netherlands are mostly unaware of the protection level of the levees, even though such protection level is extremely high.

Moreover, recent studies in Germany (Bubeck et al., 2013) and France (Poussin et al., 2014) have found that households living in protected areas can in fact take even more risk mitigation measures, or they are more likely to have flood insurance (Bubeck et al., 2013), than the ones in unprotected areas. The latter effect is caused by the set-up of the German insurance system, which highlights the importance of contextual factors on the levee effect.

## 3 Lack of knowledge

While the levee effect has been described by many authors in different parts of the world, these studies are fragmented and have used completely different methods, hampering comparative analyses. Moreover, while some scholars have focused on the evaluation of increasing exposure, such as the intense urbanisation of flood-prone areas, very few studies have focused on increased vulnerability, such as the false sense of security caused by the presence of levees. Thus, it is still unclear what the social, technical and hydrological conditions are that can (or cannot) trigger the emergence of the levee effect and to what extent. Owing to this major lack of fundamental knowledge, these effects are typically neglected in flood risk studies. This can introduce a systematic bias in the selection or prioritization of alternative strategies for flood risk reduction, for example by favouring structural measures over non-structural options like early warning systems (Pappenberger et al., 2014, precautionary measures (Kreibich et al. 2015) and relocation (Alfieri et al., 2016).

## 4 Research agenda

Hence, we call upon hydrologists, social scientists, economists, policy makers, and flood risk experts and managers to work together, and fill this gap in knowledge on the side effects of structural flood protection measures, which hinders the development of robust and sustainable strategies to reduce the negative impacts of floods. New empirical research is needed to reveal the social, technical and hydrological factors producing the levee effect, and distinguish between intended and unintended effects of structural flood protection. Our suggestion for a research agenda comprises the following three components: 1) comparative analysis of a large datasets of different case studies; 2) long-term monitoring of exposure and vulnerability dynamics; and 3) utilisation and development of new methods to explore long-term dynamics of flood risk changes and unravel the primary mechanism generating levee effects.

### 4.1 Comparative analysis

Empirical research commonly relies on specific case studies, which are unique and have their own characteristics and processes. This can make it challenging to draw general, transferable conclusions. An approach to tackle this challenge is a comparative analysis (Blöschl et al. 2013; Kreibich et al. 2017) with the aim of finding general patterns in a large set of diverse case studies in different contexts. For instance, to support universal parameter estimation for hydrological models the Model Parameter Estimation Experiment (MOPEX) assembled and analysed a large number of data sets for a wide range of river basins throughout the world (Wagener et al., 2006; Duan et al., 2006). To better understand the unintended consequences of structural flood protection, there is also a need for comparative analysis of the evolution of urban planning

and risk assessment policies, legislation and practices – including issues such as the decision making processes to define building constraints in risky areas, institutional communication strategies or the relationship between scientific and policy innovation in risk assessment. The socio-hydrological framework (Sivapalan et al., 2012), and its specific application to disaster risk reduction (Di Baldassarre et al., 2018), can provide guidance about the set of key variables to perform such a comparative analysis of the levee effect.

Hence, we suggest to identify and analyse case studies of potential or actual occurrence of the levee effect across different hydrological, technical, social, and cultural settings, and identify common patterns and factors that produce (or not) levee effects. Some examples of potential case studies across different contexts are provided in Table 1.

### 4.2. Long-term monitoring of exposure and vulnerability dynamics

Currently, the analysis of the levee effect is largely hampered by the absence of reliable long-term information on exposure and vulnerability in the focus areas. The monitoring of spatial and temporal dynamics in vulnerability is still largely missing, and strongly limited to locations that have recently experienced catastrophic flooding. Table 1 provides an overview of the types of observations needed to uncover the unfolding of levee effects, together with the actual data availability in the case studies. The table highlights that, while systematic time series of flood hazard and exposure can be more easily obtained, systematic information across decades about vulnerability is almost never available because surveys and interviews are typically performed at one point in time only, i.e. cross-sectional.

Thus, we suggest complementary empirical data collection in the case studies via longitudinal studies, where individuals, communities and decision makers are repeatedly interviewed to assess how changes in flood protection levels influence vulnerability and urban growth over time. Moreover, ideal case studies should also allow the analysis of counter-factual cases, i.e. how would risk have developed in an area had levees not been built. Such a study can be done by comparing urban growth in two adjacent areas, one protected by a levee and one which is not.

### 4.3 Exploitation of new models, concepts and data

We can draw from new approaches that have been recently developed for the study of socio-nature interactions in various interdisciplinary fields, such as ecological economics, behavioural sciences, social ecology, and sociohydrology (Folke et al., 2005; Ostrom, 2009; Kallis and Norgaard, 2010; Sivapalan et al., 2012; Montanari et al., 2013; Di Baldassarre et al., 2013a; Aerts et al., 2018). In particular, new opportunities to simulate behavioural responses to changing flood risk and flood risk management policies are offered nowadays by system dynamics (Di Baldassarre et al., 2013a) and agent-based modelling (Werner and McNamara, 2007; Aerts et al., 2018). An example of a paper that integrates behavioural theories of decision making under risk in hydrological modelling is the research by Haer et al. (2017). They apply the well-known Expected Utility Theory (Neumann and Morgenstern, 1954) of individual decision making under risk, to simulate household flood preparedness behaviour under increasing flood risk. The same study compares this behaviour with boundedly rational behaviour, using Prospect Theory developed by Tversky and Kahnemann (1979). The latter captures situations in which individuals make flood adaptation decisions, while either (under-) or overweighting (high-) low-probability flood events in

their decision to invest in flood damage mitigation measures. Another study is Haer et al. (2016) who apply an agent based model that includes decision rules based on Protection Motivation Theory to show the effect of risk communication on flood adaptation decisions.

These new models can guide empirical data collection to test alternative hypotheses about the primary mechanisms that can (or not) generate the levee effect in different contexts. This knowledge can be complemented by participatory approaches for co-generating knowledge between experts and stakeholders (Nature, 2018) in order to identify technical and policy options to address the unintended consequences of structural flood protection. Moreover, the protection motivation theory can also help explain the mitigation behaviour of individuals, which influences the vulnerability of those living behind the levees (Bubeck et al., 2012). It is particularly important to focus on what motivates protection and to provide a link between protection and communication theory/ies, by clearly identifying which communication tools and contents trigger attitudinal and behavioural change, e.g. for residual risk communication. Lastly, the increasing availability of remotely sensed data and advanced information extraction methods, such as night-light data extraction (Ceola et al., 2014; Mård et al., 2018), allows analyses of exposure dynamics over longer time spans.

## 5. Summary

We posit that exploiting these different methods, concepts and data within the suggested research agenda would significantly improve our understanding of the unintended effects of flood protection. This advanced knowledge will improve our ability to assess and explain changes in flood risk. Also, it will provide more empirical evidence supporting the selection of strategies and measures for flood risk reduction. More specifically, Table 2 shows the main research questions that remain unanswered, what elements of the proposed research agenda can help address them, and how addressing these questions can contribute to better flood risk policies.

**Acknowledgments**

This work was developed within the activities of the working group on Changes in Flood Risk of the Panta Rhei research initiative of the International Association of Hydrological Sciences (IAHS). GDB was supported by the European Research Council (ERC) within the project "HydroSocialExtremes: Uncovering the Mutual Shaping of Hydrological Extremes and Society", ERC Consolidator Grant No. 761678. KA was supported by Innovation Fund Denmark through the Water Smart Cities Project, grant 5157-0009B. PB is supported by a Royal Society Wolfson Research Merit Award and a Leverhulme Research Fellowship. PJW and WJWB received funding from the Netherlands Organisation for Scientific Research (NWO) in the form of VIDI grants 016.161.324 and 45214005.

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

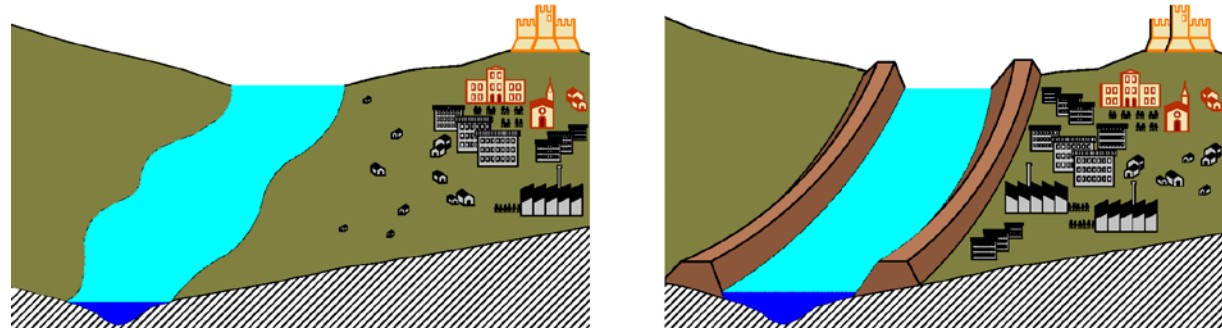

**Figure 1: Hypothetical urbanisation patterns without (left) and with (right) levees. The presence of levees often triggers more intense urbanization (in grey) in flood-prone area, which can offset (at least part of) the initial benefits of flood protection.**

**Table 1. Monitoring levee effects over time – data needs for an empirical analysis of the levee effect and their availability in different hotspots across decades**

| Data needs (ideal case study) | | | | |
|---|---|---|---|---|
| (ideally all data should be available for the same time period over several decades) | **Time series of floods** Flood information: e.g. annual maximum flows or peaks over a threshold. | **Change in flood protection standards** Data/indicators/proxies: e.g. building times and heights of levees (with some reasonable resolution e.g. 10-30 years). | **Change in flood exposure** Data/indicators/proxies: e.g. spatio-temporal changes in population density, asset values, land use in protected flood plain (with some reasonable resolution e.g. 10-30 years). | **Change in flood vulnerability** Data, indicators, or proxies: e.g. risk awareness and preparedness studies (with focus on levee effect), emergency management (e.g. early warning times), insurance cover, evolution of regulatory frameworks, legislation, policies, decision making processes and communication strategies for hazard/risk assessment. |
| **Actual data availability** | | | | |
| Dresden, Germany | Annual maximum river flows. | Available. | Land use reconstruction from 1790-2009 Estimate of asset value of residential buildings since 2000. | Survey data in Dresden: 2002: 300 households 2005/2006: 21 households 2013: 117 households |
| Cologne, Germany | Annual maximum river flows. | Available. | Development of the population since 1993 until 2020 for 80+ districts of Cologne. | Survey data from 2012 on risk perception, perceptions towards flood risk management. Can be compared to other areas that have a much higher flood risk compared with Cologne. |
| North East Italy | Annual maximum river flows. | Qualitative information available in the technical municipal and provincial offices. | Data available on: i) land use change (municipal urban plans) and construction of protection works; ii) changes in social vulnerability and population density at municipal level years (Official National Census data, conducted every 10 years since 1900). | Risk awareness and preparedness surveys conducted in 2005 (N=400, Trento area; N=176 Bolzano/Bozen area; N=100 Malborghetto Valbruna). Emergency plans and flood risk maps available. |
| The Netherlands | Annual maximum river flows. | Available. | Census data and land-use maps. | Risk awareness surveys in 2008. |
| Sacramento, USA | Annual maximum river flows. | Available. | Census data and land-use maps. | Risk awareness surveys in 2010. |
| Jamuna River floodplain in Bangladesh | Annual maximum river flows. Flood extent maps. | Available. | Census data and land-use maps. | Risk awareness surveys in 2017 |
| Denmark | Levees are for sea surges. Detailed time series, 10 series longer than 100 years. | Large flood in 1872 led to construction of large dike to protect valuable farmland. No larger change in standards since then. | National compensation scheme in place since 1980s. | Land use change and change of human preference imply that levees are protecting the wrong locations. |
| Vienna, Austria | Time series of floods. | Reports about the various projects that were undertaken throughout the years to update the flood protection system of Vienna. | Available. | No data available. |
| Calabria region, Italy | Time series of flood levels. Discharge data are not available: we deal with typically Mediterranean ungauged torrential streams. The series of maximum rainfall events can be used as a proxy of river discharge Historical series of elements damaged by floods throughout the time series | Qualitative information that can be obtained from the comparative analysis of the different types of structural works realized during the period 1820-present. | Temporal series of realisation of protection works (levees, check dams and other types) and major land transformation since 1850. Number of inhabitants obtained from Official National census: since 1900 every 10 years. Map of urbanized sectors in two or three times, depending on the availability of air photos. | Flood risk maps of PAI (Piano di Assetto Idrogeologico): these maps realized on 2000, classify territory according to four different flood risk levels. Official flood risk maps of PAI: updated version 2016. |
| Lodi, Italy | Annual maximum river flows. Annual maximum precipitation. | Executive projects of the levee system built after the 2002 flood with information about height, material, design safety level, costs and path. | Urbanization patterns (i.e. buildings construction time) since 1920 Number of inhabitants from official national census since: since 1900 every year Orthoimages: since 1950. | Risk awareness and preparedness survey of people affected in 2002 and still living in the area (10 households ongoing). Emergency plans and flood risk maps available. |

**Table 2. Summary of the research agenda in terms of questions, methods, and outcomes**

| Research question | Methods | Potential outcomes |
|---|---|---|
| Which socio-hydrological factors enhance or alleviate the levee effect? | Comparative analysis (Section 4.1) | These factors would enable the identifications of contexts in which increasing structural protection levels can be less beneficial than expected. Hence, it will contribute to a better development of risk reduction policies. |
| How does structural flood protection influence changes in risk perception and flood preparedness decisions? | Longitudinal surveys (Section 4.2) | Understanding what influences changes in risk perception and flood preparedness decisions would suggest how to improve risk awareness campaigns, thus alleviating the levee effect. |
| How does structural flood protection influence changes in human settlements? | Long-term monitoring (Section 4.2) | Understanding how flood protection shapes human settlements would support a more realistic assessment of long-term (decadal) changes in flood exposure. |
| How does flood risk changes over time in differ contexts, e.g. with/without structural flood protection? | System dynamic modelling, agent based modelling and new datasets (Section 4.3) | Modelling or observing behavioural responses would support a more realistic assessment of long-term changes in flood risk in different contexts. |