# Peer review of "Hess Opinions: An interdisciplinary research agenda to explore the unintended consequences of structural flood protection"

_Hydrology and Earth System Sciences, 2018_

## Referee Comment (RC1) · Anonymous Referee #1 · 7 Aug 2018

The paper by Di Baldassare et al. aims to review and provide critical analysis of the incidental impacts associated with structural flood protection measures. The authors make the case that the unintended consequences, often referred to as the 'levee effect' (among other names), are ubiquitous but poorly understood due to a i) lack of generalizable knowledge or comparative analysis, ii) dearth of holistic and long-term data and monitoring, and iii) incomplete or insufficient methods. While the levee effect phenomena has been established in the literature for decades, a coherent, interdisciplinary research agenda has not been set to understand the causes, feedback dynamics, and vulnerabilities of these systems. This paper provides such an agenda, albeit very briefly. In general, the commentary is well-written, of good quality, and adequately

addresses the aims set forth. Therefore, I recommend the paper be formally accepted for publication upon the following improvements.

Specific comments 1) Identify specific research questions that remain outstanding and how your research agenda will help address them. The language expressing the current knowledge gap is vague. One of the key purposes of these paper is to set a research agenda for the field; as such, this paper would be greatly strengthened by i) providing specific research questions that remain unanswered, ii) explaining how answering these questions will lead to transformative results, and iii) how the proposed research agenda will enable these questions to be addressed. Not only will this help the authors meet one of their chief aims, but it has the potential to make a true novel contribution to the literature, possibly serving as a catalyst for further research in this area. 2) Make the two examples more explicit. In the abstract, the authors say their commentary explores the "intended benefits and unintended effects of flood protection with two main examples". However, these two examples are difficult to identify given the authors briefly highlight numerous examples throughout their commentary.

Technical corrections Be consistent in use of Oxford comma.

Page 1 Line 32: "phenomenon, by" -> "phenomenon by" Line 32: "effects of flood protection" -> "effects of structural flood protection" Line 33: "and then propose" -> "and then we propose"

Page 2 Line 18: "2 The troubles with flood protection" -> "2 The troubles with structural flood protection" Line 25: "Sacramento valley" -> "Sacramento Valley" Line 26: "and in the United States" -> "in the United States"

Page 3 Line 5-6: "transport cost, or areas in cities to benefit" -> "transport cost or areas in cities that benefit" Line 16: "not always realistic, while large" -> "not always realistic. Moreover, large" Line 19: remove "for instance" Line 24: "flooding" -> "flood"

Page 4 Specify the meaning of the Likert scale values used in the survey (e.g. a

response of 1 signifies strong disagreement with the statement, while a response of 5 indicates strong agreement).

Page 6 Line 18: "together, within" -> "together within" Line 20: "flood risk, and provide" -> "flood risk and provide"

Table 1: "evolution of of regulatory" -> "evolution of regulatory"

---

## Referee Comment (RC2) · Anonymous Referee #2 · 11 Aug 2018

This is an interesting work based on deep insights and years of research on the topic by the authors. I think that the paper can be improved further by addressing the following points:

1) Page 2, Lines 7-8: "This study made the (common) assumption that future flood exposure depends on socioeconomic trends only, and not on the level of flood protection." Can you elaborate more on the difference between these socioeconomic trends and the trend of more intense urbanization of flood-prone areas behind the levee? Isn't the latter a subset of the former? That is, aren't increased economic activities on flood-plains part of greater socioeconomic trends that you're referring to? This is unclear to

me.

2) Page 2, Lines 26-31: Here, you describe a case, talking about things such as how a flooding event in 1953 reduced population density of floodplain. However, it wasn't clear to me initially where this case is based on and what is the context of this 1953 flooding. I had to search other parts of the paper to find out that this is based on the Netherlands. It will be more reader-friendly if you can clearly mention that this flooding and social changes on floodplain are based on the Netherlands.

3) Page 3, Lines 1-10: Here, you contrast two cases: "urban growth behind the dikes is often factored into the risk analysis" and "urban growth in flood-prone areas goes beyond original plans, potentially leading to unforeseen increase in flood risk" (thus, leading to more levee development). This is interesting. But it wasn't clear to me how the two cases differ in terms of flood protection level. In the former case when risk analysis includes urban growth behind the levees, are the levees built much higher in the first place to reflect this expected growth? How will long-term flood protection level and flood risk be different between these two cases?

4) Page 3, Lines 19-21: This transference of risk to downstream due to hydraulic inter-actions stemming from heightening of upstream levees is interesting. Readers might benefit from little more discussion on this.

5) Pages 5: Comparative analysis is identified as a future challenge. I was surprised to see that the authors don't talk about how a framework can help with the task. In order to compare different cases in a consistent and structured way, a framework is needed. A framework defines a general set of variables and their potential relationships that an analyst should consider when examining cases.

6) Page 6: The authors talk about how new methods, concepts, and data can help advance sociohydrology research. It will be more helpful if the authors describe how such multiple methods work together and create synergies. Also, no methods of behavioral sciences are really described here. Modeling behavioral response using differential

equation is not really a behavioral science method.

7) Page 6, Line 12: I think that Aerts (2018) is a review paper, not ABM paper.

---

## Short Comment (SC1) · 30 Aug 2018

The authors discuss a very interesting topic (the unintended consequences of structural flood measures) and propose a good agenda. I consider it especially important because few weeks ago social media were flooded with comments praising the Tokyo underground flood tunnels, despite some Japanese researchers note that such ultra expensive infrastructure may lead to a false sense of security. Moreover, Matsuda (2013) suggests that the vicious cycle (large investment -> sense of security -> concentration of property and people -> increased potential for damage -> large investment -> . . .) has been one of the great problems/challenges of Tokyo since the 19th century.

The proposed agenda is a very interesting and important topic. Nevertheless, I believe the current agenda has some limitations. Thus, I would like to suggest some points that may be included:

In 4.1 Comparative analysis:

Future conditions. I would like to stress the importance simulate the effects (hazards) of future conditions. Regardless the changes in vulnerability, exposure or resilience, the hydrological drivers and the hydrodynamic conditions of future flood events will be different. It would be useless to analyze future vulnerability changes without considering future conditions and future hazards. Climate change and economic growth are key factor that will alter the hydrological response of catchments. For instance, Moya Quiroga et al., (2016) suggests that future hydrological design discharges may increase up to 15% due to climate change, while Winsemius et al., (2015) suggests that flood contribution by climate change may be small compared with economic growth.

Worldwide case studies. Most of the potential case studies are from Europe (only one case from America). European basins are small ones, especially when compared with basins from South America (SA) or South East Asia (SEA) (smaller size, smaller peak flows, hydrographs and less sediments). Besides, as SA and SEA basin are on tropical locations, they are more sensitive to future changes. Thus, it would be important to include more world case studies.

Additional infrastructure. The presented agenda focuses on levees. It would be important to analyze additional infrastructure such as dams or roads. Although the objective of such infrastructure (roads, dams) is not flood protection, they are designed based on given hydrological condition that will be changed and have not been analyzed. Moreover, the upstream effects of protection infrastructure has not been analyzed.

Upstream consequences. Upstream consequences of dams and levees have always been neglected. Only recently new studies analyzed some upstream consequences of dams. For instance,, new studies revealed that Three Gorges Dam would induce more

frequent impounding periods with higher risks of risks of infection and illness stopping transport of pollutants, higher pollution in upstream tributaries (Sha et al., 2015; XIao et al., 2013) and more.

In 4.3 Exploitation of new methods, concepts and data

Currently there is lack of knowledge regarding the cascade effects. Flood structural protection measures are designed to keep and store the water on the upstream. Such ponded water usually becomes a breeding pool for several infections and diseases. For instance, ponded water in floodplains and reservoirs usually becomes a breeding pool for mosquitoes; hence, are likely to increase the transmission of vector borne diseases like malaria (Endo and Eltahir, 2018; Moya Quiroga et al., 2018).

References

Endo, N. Eltahir, E. 2018. Modelling and observing the role of wind in Anopheles population dynamics around a reservoir, Malaria journal,

Xiao, G., Qiu, Z., Qi, J., et al., 2013. Occurrence and potential health risk of Cryptosporidium and Giardia in the Three Gorges Reservoir, China, Water Research, 47 (7), 2431-2445.

Sha, Y., Wei, Y., Li, W., et al., 2015. Artificial tide generation and its effects on the water environment in the backwater of Three Gorges Reservoir, Journal of hydrology, 528, 230-237.

Matsuda, I. 2013. Verifying Vulnerability to Natural Disasters in Tokyo, Journal of Geography, 122 (6), 1070-1087

Moya Quiroga, V., Kure, S., Udo, K., Mano, A. 2018. Analysis of exposure to vector borne diseases due to flood duration for a more complete flood hazard assessment: Llanos de Moxos - Bolivia, Revista Iberoamericana del Agua RIBAGUA, 5(1), 48-62. DOI 23863781.2017.1332816.
Moya Quiroga, V., Kure, S., Udo, K., Mano, A. 2016. Changes in the hydrological design discharges due to Climate Change: Bolivian Amazonia, Journal of Environmental Systems and Engineering, Japan Society of Civil Engineers Series G (JSCE-G), 72(5), I247-I252. http://doi.org/10.2208/jscejer.72.I_247

Winsemius, H., Aerts, J., van Beek. L., et al. 2015. Global drivers of future river flood risk, Nature Climate Change, 6(4), 1-5.
* * *

---

## Author Comment (AC1) · 10 Sep 2018

We thank Anonymous Referee #1 for her/his positive comments about our opinion paper, and for providing useful and constructive comments.

We will carefully revise our manuscript and address all the points raised by the Referee. More specifically, we will improve the identification of research questions as suggested by the Referee (point number 1), and make our examples more explicit (point number 2). We will also make a better link between the text and Figure 1, to clarify the Dutch example.

[Figure]

We also thank the Referee for providing technical corrections. We will consider all of them while revising our manuscript. In particular, the quoted reference to "two examples" in the abstract was a typo. It should have been "the examples".

---

## Author Comment (AC2) · 10 Sep 2018

We thank Anonymous Referee #2 for her/his positive comments about our opinion paper, and for providing useful and constructive comments.

We will carefully revise the manuscript and address all the points raised by the Referee:

1) As suggested, we will clarify in the revised manuscript the difference between socioeconomic trends and more intense urbanization of flood-prone areas behind the levee. This will also be done by better linking the text with Figure 1.

2) The Referee is right. The revised manuscript will make clear that the text is about

the Dutch example.

3) This difference will be clarified in the revised text. As for point number 1, a better link between the text and Figure 1 will help to make this clearer.

4) As suggested, more discussion about the risk being transferred downstream will be included in the revised manuscript.

5) The socio-hydrological framework can help in carrying out comparative analyses. This will be clarified in the revised manuscript.

6) The revised manuscript will describe how different methods work together and create synergies, as suggested by the Referee. Indeed, behavioural science methods are not described, but references will be added in the revised manuscript.

7) Reference to ABM papers will be added to the revised manuscript.

———————————————————

---

## Author Comment (AC3) · 10 Sep 2018

We thank Dr. Moya Quiroga for the positive comments about our opinion paper and for providing useful and constructive comments that can help enrich our commentary. We will consider all these aspects during the revision process.

We also thank Dr. Moya Quiroga for suggesting a number of references. Many of them are very relevant. For example, the points made about: i) the vicious cycle with the Tokyo example (Matsuda, 2013), and ii) the comparison of changes in flood risk and economic growth (Winsemius et al, 2015) are definitely in line with our opinion piece.

---

## Author Response (AR1)

**Editor**

*The referees found your paper significant and well-written, and I agree with their recommendation of minor revisions. Please ensure that you address the suggestions in the short comment, which I found to be very useful (particularly the point to extend your analysis beyond the European emphasis to more worldwide examples).*

We thank the editor for handling our paper. We have addressed all the referees' suggestions, which helped us improve this opinion piece. This letter provides a point-by-point response to all comments. To go beyond the European emphasis, the revised manuscript includes references to the Sacramento Valley and the case of New Orleans in USA as well as flood protection in Bangladesh (with a new reference to the paper recently published by Ferdous et al., in HESS).

**Anonymous Referee #1**

*The paper by Di Baldassare et al. aims to review and provide critical analysis of the incidental impacts associated with structural flood protection measures. The authors make the case that the unintended consequences, often referred to as the 'levee effect' (among other names), are ubiquitous but poorly understood due to a i) lack of generalizable knowledge or comparative analysis, ii) dearth of holistic and long-term data and monitoring, and iii) incomplete or insufficient methods. While the levee effect phenomena has been established in the literature for decades, a coherent, interdisciplinary research agenda has not been set to understand the causes, feedback dynamics, and vulnerabilities of these systems. This paper provides such an agenda, albeit very briefly. In general, the commentary is well-written, of good quality, and adequately addresses the aims set forth. Therefore, I recommend the paper be formally accepted for publication upon the following improvements.*

We thank the Referee#1 for being positive about our paper. We have addressed all her/his specific points as specified below.

*1) Identify specific research questions that remain outstanding and how your research agenda will help address them. The language expressing the current knowledge gap is vague. One of the key purposes of these paper is to set a research agenda for the field; as such, this paper would be greatly strengthened by i) providing specific research questions that remain unanswered, ii) explaining how answering these questions will lead to transformative results, and iii) how the proposed research agenda will enable these questions to be addressed. Not only will this help the authors meet one of their chief aims, but it has the potential to make a true novel contribution to the literature, possibly serving as a catalyst for further research in this area.*

As suggested by the Referee, the revised manuscript has a new table (Table 2) with three columns that i) provides specific research questions that remain unanswered, ii) explains how answering these questions will lead to transformative results, and iii) what methods in the proposed research agenda will help these questions to be addressed. See Revised Manuscript with track change (Section 5 and Table 2).

**Table 2. Summary of the research agenda in terms of questions, methods, and outcomes**

| Research question | Methods | Potential outcomes |
|---|---|---|
| Which socio-hydrological factors enhance or alleviate the levee effect? | Comparative analysis (Section 4.1) | These factors would enable the identifications of contexts in which increasing structural protection levels can be less beneficial than expected. Hence, it will contribute to a better development of risk reduction policies. |
| How does structural flood protection influence changes in risk perception and flood preparedness decisions? | Longitudinal surveys (Section 4.2) | Understanding what influences changes in risk perception and flood preparedness decisions would suggest how to improve risk awareness campaigns, thus alleviating the levee effect. |
| How does structural flood protection influence changes in human settlements? | Long-term monitoring (Section 4.2) | Understanding how flood protection shapes human settlements would support a more realistic assessment of long-term (decadal) changes in flood exposure. |
| How does flood risk changes over time in differ contexts, e.g. with/without structural flood protection? | System dynamic modelling, agent based modelling and new datasets (Section 4.3) | Modelling or observing behavioural responses would support a more realistic assessment of long-term changes in flood risk in different contexts. |

*2) Make the two examples more explicit. In the abstract, the authors say their commentary explores the "intended benefits and unintended effects of flood protection with two main examples". However, these two examples are difficult to identify given the authors briefly highlight numerous examples throughout their commentary.*

There is a misunderstanding here due to an incorrect use of the term "two". The text of the abstract was revised. See Revised Manuscript with track change.

*Technical corrections Be consistent in use of Oxford comma. Page 1 Line 32: "phenomenon, by" -> "phenomenon by" Line 32: "effects of flood protection" -> "effects of structural flood protection" Line 33: "and then propose" ->"and then we propose"*

Amended.

*Page 2 Line 18: "2 The troubles with flood protection" -> "2 The troubles with structural flood protection" Line 25: "Sacramento valley" -> "Sacramento Valley" Line 26: "and in the United States" -> "in the United States"*

Amended.

*Page 3 Line 5-6: "transport cost, or areas in cities to benefit" -> "transport cost or areas in cities that benefit" Line 16: "not always realistic, while large" -> "not always realistic. Moreover, large" Line 19: remove "for instance" Line 24: "flooding" -> "flood"*

Amended.

*Page 4 Specify the meaning of the Likert scale values used in the survey (e.g. a response of 1 signifies strong disagreement with the statement, while a response of 5 indicates strong agreement).*

Amended.

*Page 6 Line 18: "together, within" -> "together within" Line 20: "flood risk, and provide" -> "flood risk and provide"*

Amended.

*Table 1: "evolution of of regulatory" -> "evolution of regulatory"*

Amended.

**Anonymous Referee #2**

*This is an interesting work based on deep insights and years of research on the topic by the authors.*

We thank the Referee#2 for being positive about our paper. We have addressed all her/his specific points as specified below.

*1) Page 2, Lines 7-8: "This study made the (common) assumption that future flood exposure depends on socioeconomic trends only, and not on the level of flood protection." Can you elaborate more on the difference between these socioeconomic trends and the trend of more intense urbanization of flood-prone areas behind the levee? Isn't the latter a subset of the former? That is, aren't increased economic activities on floodplains part of greater socioeconomic trends that you're referring to? This is unclear to me.*

As suggested, the revised manuscript clarifies the difference between socioeconomic trends and more intense urbanization of flood-prone areas behind the levee. This is done by better linking the text with Figure 1. See Revised Manuscript end of Section 1.

*2) Page 2, Lines 26-31: Here, you describe a case, talking about things such as how a flooding event in 1953 reduced population density of floodplain. However, it wasn't clear to me initially where this case is based on and what is the context of this 1953 flooding. I had to search other parts of the paper to find out that this is based on the Netherlands. It will be more reader-friendly if you can clearly mention that this flooding and social changes on floodplain are based on the Netherlands.*

The Referee is right. This was made clear in the revised manuscript.

*3) Page 3, Lines 1-10: Here, you contrast two cases: "urban growth behind the dike 3) Page 3, Lines 1-10: Here, you contrast two cases: "urban growth behind the dikes is often factored into the risk analysis" and "urban growth in flood-prone areas goes beyond original plans, potentially leading to unforeseen increase in flood risk" (thus, leading to more levee development). This is interesting. But it wasn't clear to me how the two cases differ in terms of flood protection level. In the former case when risk analysis includes urban growth behind the levees, are the levees built much higher in the first place to reflect this expected growth? How will long-term flood protection level and flood risk be different between these two cases?*

As suggested, the revised manuscript clarifies how urban growth is typically factored. Also in this case, a better link to Figure 1 was provided. See Revised Manuscript.

*4) Page 3, Lines 19-21: This transference of risk to downstream due to hydraulic interactions stemming from heightening of upstream levees is interesting. Readers might benefit from little more discussion on this.*

As suggested, we expanded this part with one reference to a paper about levee wars. See revised manuscript.

*5) Pages 5: Comparative analysis is identified as a future challenge. I was surprised to see that the authors don't talk about how a framework can help with the task. In order to compare different cases in a consistent and structured way, a framework is needed. A framework defines a general set of variables and their potential relationships that an analyst should consider when examining cases.*

We agree with the Referee. We added references to the socio-hydrological framework and its application to disaster risk reduction for guiding comparative analyses.

*6) Page 6: The authors talk about how new methods, concepts, and data can help advance sociohydrology research. It will be more helpful if the authors describe how such multiple methods work together and create synergies. Also, no methods of behavioural sciences are really described here. Modeling behavioral response using differential equation is not really a behavioural science method.*

The Refereer is correct, and we have revised that section by referring to some important behavioural theories that were recently integrated in coupled 'hydrologic-agent based' models. See revised manuscript with track change Section 4.3.

The revised manuscript also clarify how multiple experts and stakeholders can work together and co-produce knowledge in the field (with reference to the last issue of *Nature* that matches perfectly our point here). See revised manuscript with track change Section 4.3.

*7) Page 6, Line 12: I think that Aerts (2018) is a review paper, not ABM paper.*

Aerts (2018) discusses the potential of ABM in integrating human behaviours into risk assessment. As suggested by the Referee, specific reference to a more specific example of ABM (human-landscape interactions and flooding in New Orleans; Werner and McNamara, 2007) was added to the revised manuscript.

[revised manuscript text omitted]